# FPGA-Based Vehicle Detection and Tracking Accelerator

**DOI:** 10.3390/s23042208

**Published:** 2023-02-16

**Authors:** Jiaqi Zhai, Bin Li, Shunsen Lv, Qinglei Zhou

**Affiliations:** 1School of Computer and Artificial Intelligence, Zhengzhou University, Zhengzhou 450001, China; 2Henan Key Laboratory of Network Cryptography Technology, Zhengzhou 450001, China

**Keywords:** FPGA, vehicle detection, accelerator architecture, YOLO, DeepSort

## Abstract

A convolutional neural network-based multiobject detection and tracking algorithm can be applied to vehicle detection and traffic flow statistics, thus enabling smart transportation. Aiming at the problems of the high computational complexity of multiobject detection and tracking algorithms, a large number of model parameters, and difficulty in achieving high throughput with a low power consumption in edge devices, we design and implement a low-power, low-latency, high-precision, and configurable vehicle detector based on a field programmable gate array (FPGA) with YOLOv3 (You-Only-Look-Once-version3), YOLOv3-tiny CNNs (Convolutional Neural Networks), and the Deepsort algorithm. First, we use a dynamic threshold structured pruning method based on a scaling factor to significantly compress the detection model size on the premise that the accuracy does not decrease. Second, a dynamic 16-bit fixed-point quantization algorithm is used to quantify the network parameters to reduce the memory occupation of the network model. Furthermore, we generate a reidentification (RE-ID) dataset from the UA-DETRAC dataset and train the appearance feature extraction network on the Deepsort algorithm to improve the vehicles’ tracking performance. Finally, we implement hardware optimization techniques such as memory interlayer multiplexing, parameter rearrangement, ping-pong buffering, multichannel transfer, pipelining, Im2col+GEMM, and Winograd algorithms to improve resource utilization and computational efficiency. The experimental results demonstrate that the compressed YOLOv3 and YOLOv3-tiny network models decrease in size by 85.7% and 98.2%, respectively. The dual-module parallel acceleration meets the demand of the 6-way parallel video stream vehicle detection with the peak throughput at 168.72 fps.

## 1. Introduction

As urbanization accelerates, new infrastructure is empowering the development of smart cities while providing new impetus to the postepidemic economy. Satisfying the need for smooth traffic flow and improving traffic efficiency (reducing the number of vehicles on the road and increasing vehicle occupancy) are the goals of intelligent transportation systems. To achieve these goals, Bilal et al. [1] used Hadoop and Spark to design a model that analyzes transportation data in real time to publish road traffic conditions for citizens in real time; Lin et al. [2] proposed a public vehicle system with edge computing to improve traffic efficiency by arranging ride-sharing among travelers; Wang et al. [3] proposed a rerouting system to help drivers make the most appropriate choice of the next road to avoid congestion; Tseng et al. [4] proposed a dynamic rerouting strategy based on real-time traffic information and decisive weights to effectively solve the road congestion problem. These studies provide an intelligent decision basis for intelligent transportation from various perspectives, such as road command and dispatch, traffic signal control changes, and traffic guidance, which are all based on traffic information in real situations.

Many detection algorithms have emerged for traffic information collection. Traditional detection algorithms such as optical flow [5] and background modeling [6] have a high algorithmic complexity and are difficult to accelerate using parallel devices, thus resulting in poor real-time detection. Methods based on the collaborative deployment of multiple sensors, such as LiDAR, GPS, and camera devices for vehicle detection are better, but deployment is costly and influenced by environmental factors [7,8]. With the advanced development of convolutional neural networks, network architectures such as R-CNN (region-based convolutional neural network) [9], Faster-RCNN (faster region-based convolutional neural network) [10], SSD (single shot multibox detector) [11], and YOLO [12] and other new network architectures have emerged and are applied to image classification, object detection, and other fields [13]. Compared with two-stage networks such as R-CNN and Faster-RCNN, the single-stage YOLO series of networks treats object detection as a regression problem to solve; thus, it has higher detection speed [14].

FPGAs are semicustom circuits that have a lower latency and higher parallelism capability than CPUs and have a lower power consumption and lower cost than GPUs. Compared to ASIC, they have a shorter design cycle, are more iterable, and are less costly. With the rapid development of deep learning frameworks, FPGAs are the best platform for the deep learning model forwarding inference acceleration [14].

There are still many challenges in FPGA hardware acceleration. The performance of hardware acceleration is directly related to the on-chip resources of FPGAs. How to use limited hardware resources to design an efficient hardware acceleration architecture is a very important research problem. For the application of the YOLO series neural network, which is computationally intensive and has huge parameters, a high memory access frequency, and complex control logic, improving the acceleration performance involves two difficulties: optimizing the computing process and optimizing the memory exchange. In order to improve the accuracy and quantity of vehicle detection, networks with higher accuracy are needed. However, due to the huge amount of network parameters and computation, these networks will bring high resource and computation costs [15]. How to reduce the parameter amount and algorithm complexity to improve the acceleration performance becomes one of the difficulties. In addition, in the process of data exchange between the on-chip and off-chip, due to the lack of the effective organization of data, the utilization of bandwidth resources is insufficient and the efficiency of parallel reading and writing is low, thus becoming a bottleneck that restricts efficient computing [16]. Optimizing the data organization and memory exchange strategy to reduce the communication cost between the on-chip storage and off-chip memory is another optimization route.

There have been many studies focusing on the FPGA-based convolutional neural network acceleration. However, the research of transplantation optimization for large models remains in theoretical research and fails to combine with specific scenarios for application deployment. Therefore, taking vehicle tracking and counting as the specific application scenario, we study the deployment of the YOLO series network acceleration on the FPGA side from the two aspects: the neural network compression and hardware accelerator design. In addition, we retrain the appearance feature extraction network of the Deepsort algorithm based on the self-generated dataset for vehicle tracking. The main contributions of this work are summarized as follows:We trained the YOLOv3 and YOLOv3-tiny networks using the UA-DETRAC dataset [17]. Then, we incorporate the dynamic threshold structured pruning strategy based on binary search and the dynamic INT16 fixed-point quantization algorithm to compress the model.A reidentification dataset was generated based on the UA-DETRAC dataset and used to train the appearance feature extraction network of the Deepsort algorithm with a modified input size to improve the vehicle tracking performance.We designed and implemented a vehicle detector based on an FPGA using high level synthesis (HLS) technology. At the hardware level, optimization techniques such as the Im2col+GEMM and Winograd algorithms, parameter rearrangement, and multichannel transmission are adopted to improve the computational throughput and balance the resource occupancy and power consumption. Compared with the other related work, vehicle detection performance with higher precision and higher throughput is realized with lower power consumption.Our design adopts a loosely coupled architecture, which can flexibly switch between the two detection models by changing the memory management module, optimizing the balance between the software flexibility and high computing efficiency of the dedicated chips.

The rest of this paper is organized as follows: Section 1 reviews the background knowledge and related work on the simplification of deep neural networks (DNNs) and the convolutional neural network acceleration based on FPGAs. Section 2 introduces our strategies of neural network compression and accelerator optimization. Section 3 presents our experiments and analysis. Finally, we conclude the paper in Section 4.

## 2. Background and Related Work

### 2.1. YOLO

YOLO was proposed by Joseph Redmon et al. in 2016. Compared with other networks in the YOLO series, YOLOv3 and YOLOv3-tiny achieves a good balance in accuracy and detection speed, and is easy to transplant to hardware.

Figure 1 shows the network structure of YOLOv3-tiny. The network has a total of 24 layers, including 13 convolutional layers, 6 max-pooling layers, 2 routing layers, 1 upsampling layer, and 2 output layers. The size of the convolution kernel of the convolution layer is 3×3 and 1×1. The model input uses RGB images with a size of 416×416×3. The prediction branches are predicted using two scales, 13×13 and 2626.

The network structure of YOLOv3 is similar to that of YOLOv3-tiny, except that it does not use max-pooling layers and uses the residual network, thereby constructing DarkNet-53 as the backbone with deeper network layers and one more prediction branch than YOLOv3-tiny, with a scale of 52×52.

### 2.2. Deepsort

Deepsort [18] is an online multitarget tracking algorithm. It considers both the detection frame parameters of the detection result and the appearance information of the tracked object, combining the relevant information of the previous frame and the current frame for prediction without considering the whole video at the time of detection. In the first frame of the video to be detected, a unique track ID is assigned to the detection frame of each target. Then, the detection object in the new frame is associated with the previously tracked object using the Hungarian algorithm [19] to obtain a global minimum of the assignment cost function. The cost function contains the spatial Mahalanobis distance [20] d(1), which measures the difference between the detected frame and the position predicted based on the previously known position of the object, and a visual distance d(2), which measures the difference between the appearance of the currently detected object and the previous appearance of the object. The cost function for assigning the detected object *j* to track *i* is shown in (1), the spatial martingale distance is shown in (2), and the visual distance is shown in (3). The meanings of the parameters in the formula are shown in Table 1.
(1)$$C_{i,j}=\lambda d^{\left(1\right)}\left(i,j\right)+\left(1-\lambda \right)d^{\left(2\right)}\left(i,j\right).$$
(2)$$d^{\left(1\right)}\left(i,j\right)={\left(d_{j}-y_{i}\right)}^{T}S_{i}^{-1}\left(d_{j}-y_{i}\right).$$
(3)$$d^{\left(2\right)}\left(i,j\right)=min\left(1-r_{j}^{T}r_{k}^{\left(i\right)}|r_{k}^{\left(i\right)}\in R_{i}\right).$$

### 2.3. Simplification of the DNN

Neural networks used in object detection, such as vehicle detection, usually require deeper network layers and more complex network structures. Therefore, such large-scale parameters and high computational complexity pose a challenge to the end-to-end deployment of DNNs.

Many studies have proposed various methods for neural network compression and simplification, including network pruning, weight quantization, approximated computing and low-rank decomposition of the weight matrix. Han et al. [21] first proposed fine-grained neural network pruning based on the finding demonstrating that removing weight parameters with neural network median values close to 0 does not affect the network performance. Such unstructured pruning granularity for a single neuron generates many sparse matrices. However, sparse matrix operations cannot use the existing mature BLAS library to obtain additional performance gains. To keep the model in a regular computational pattern after pruning, attention has been given to structured pruning, which achieves compression by filtering out the channels and convolution kernels below a certain threshold according to a custom condition. Hao et al. [22] proposed a method for pruning filters by estimating the ability of filters to affect the network using the L1 paradigm, but the pruning caused a large loss in detection accuracy. Yang et al. [23] treated the determination of the pruning ratio as a search problem and used a search algorithm to calculate the pruning ratio for each layer as a way to reduce the loss due to pruning. Liu et al. [24,25] proposed and improved a channel pruning method based on sparse training that compresses the coefficients of the batch normalization (BN) layer by sparse training and then determines and removes the unimportant channels according to the coefficient. After the pruning is completed, the model is fine-tuned to recover the accuracy, which greatly reduces the model parameter redundancy while minimizing the loss. Qiu et al. [26] proposed a dynamic precision data quantization method to quantify the weight values, bias values, and intermediate results into fixed points of different precision. Cardarilli et al. [27] showed that using approximate computing techniques can reduce power consumption in neural networks.

### 2.4. CNN Accelerator Based on an FPGA

There has been much research focusing on the FPGA-based acceleration of convolutional neural networks. Ma et al. [28] improved the performance of the accelerator by performing quantitative analysis and optimizing CNN loops to reduce memory access and data exchange, but their accelerator did not validate the acceleration performance for large networks. Zhang et al. [29] proposed optimizing the on-chip cache using ping-pong operations to hide the data transfer latency and designed it to search the accelerator optimization space using the roofline model. However, they only designed the hardware architecture. Lu et al. [30] first used the Winograd algorithm in CNN operations to reduce the convolutional computational complexity and proposed row buffers to achieve efficient data reuse. Later, in the literature [31], the Winograd algorithm was proposed to be combined with the CNN sparsity to improve accelerator performance, but the model used in its evaluation was simple. Bao et al. [32] used a fixed-point quantization approach to reduce FPGA resource consumption and proposed a buffer pipeline approach to further improve the accelerator efficiency while reducing the resource and power overhead. Wang et al. [33] introduced a new unstructured sparse convolution algorithm using a lower quantization method and an end-to-end design space search sparse convolution dedicated circuit architecture, which achieved high computational efficiency, but its performance-to-power ratio was relatively low.

The above studies have made great contributions to deploying AI directly on edge devices, but as the models become more complex, research on the optimization of large models remains in theoretical studies and fails to be deployed in conjunction with specific scenario applications.

The neural network acceleration contains complex operators and memory management modules, so using HDL (Hardware Description Language) to directly describe the framework has a long development cycle, making it difficult to explore the design space. HLS uses C/C++ to describe the framework from a high level. It greatly improves development efficiency due to the rapid conversion of high-level code to FPGA implementation [34]. Many studies on neural network acceleration have been implemented based on HLS, and the HLS tools for neural network acceleration have been improved and expanded to make development easier and faster. We designed and implemented a vehicle detector based on an FPGA using HLS.

## 3. Optimization and Implementation of the Vehicle Detector

Combined with the background knowledge in Section 1, when designing the vehicle detector, we use the special vehicle detection dataset UA-DETRAC to conduct basic training for the models. Then, we use a binary search-based dynamic threshold structured pruning method to reduce the number of model channels. After pruning, the model was quantified into 16-bit fixed points. Meanwhile, we generate a vehicle tracking dataset REID-UA-DETRAC from the UA-DETRAC dataset and use it to train the appearance feature extraction network in the Deepsort algorithm to improve the tracking performance on vehicles. In terms of hardware optimization, we improve the detection speed of video streams from the perspectives of optimizing the memory transmission and algorithm complexity.

### 3.1. Model Compression

#### 3.1.1. Structured Pruning Based on Dynamic Threshold of Binary Search

Model pruning is still the most effective method of neural network compression thus far, and it reduces the storage space and inference time required for the model by eliminating parameters from the model that have little impact on the detection effect [35].

The workflow of the structured pruning based on the dynamic threshold of the binary search adopted in this paper is shown in Figure 2. It can be described as multiple iterations of the following process on the base-trained model. Sparse training is first performed using a sparse regularization algorithm. Then, we determine the max pruning threshold based on the distribution of scale factors to remove channels whose contribution values are less than the threshold. In order to compress the model as much as possible, we used the binary search method when determining the pruning threshold in each round, and performed multiple iterations with 50% as the starting point to achieve the best balance between pruning rate and accuracy. Finally, the knowledge distillation strategy is used to fine-tune the network accuracy.

Sparse regularization training first introduces a scaling factor for each channel, which is used to multiply with the output of that channel. The scaling factors are trained jointly with the network weights and are sparsely regularized during the training to identify insignificant channels. The objective function of the sparse regularization training is shown in (4), where (x,y) represents the input and target of training and *W* represents the trainable weight. The first term represents CNN training losses, g(·) is the sparse penalty function for the scaling factor, and g(s)=|s|. λ is used to balance the effect of two terms as the result. We use the subgradient descent algorithm to optimize the nonsmooth L1 penalty term.
(4)$$L=\sum_{\left(x,y\right)}l(f(x,W),y)+\lambda \sum_{\gamma \in \Gamma }g(\gamma ).$$

The structure of the network after sparse regularization training is shown in Figure 3a. We prune the channels whose contribution value is less than the threshold value to obtain the network structure, as shown in Figure 3b.

#### 3.1.2. Dynamic 16-bit Fixed-Point Quantization

When the model is trained in the GPU, it usually uses a 32-bit floating point to express the weight, gradient, and activation value of the network. Using floating-point operations increases the overhead of the computational unit; thus, currently, lower bit-width fixed-point numbers are usually used for the inference process of neural networks. We carried out the dynamic 16-bit fixed-point quantization for weight bias parameters, feature mapping, and intermediate results in three quantization stages [36]. The conversion of fixed-point and floating-point numbers is shown in Equations (5) and (6), where $x_{q}$ is a fixed-point number, *x* is a floating-point number, and *Q* is used to specify the base point position of the fixed-point number.
(5)$$x_{q}=\left(int\right)x*2^{Q}.$$
(6)$$x=\left(float\right)x_{q}*2^{-Q}.$$

For a given fixed-point number, its actual floating-point value is a function of the bit width *w* and the exponent character *Q*, as shown below:(7)$$V_{fixed}=f\left(w,Q\right)=\sum_{i=0}^{w-1}B_{i}\cdot 2^{-Q}\cdot 2^{i},B_{i}\in 0,1.$$

In the stage of quantifying the weight values and bias values, the optimal *Q* values are analyzed for each layer dynamically using the approach shown in Equations (8) and (9), so that the absolute error sum of the original value of the weight bias and the quantized value is minimized. $W_{float}^{l}$ and $b_{float}^{l}$ are 32-bit floating-point values of the *l*-th layer weights and biases, respectively, and $W_{fixed}^{l}\left(w,Q\right)$ and $b_{fixed}^{l}\left(w,Q\right)$ are 16-bit fixed-point values of the *l*-th layer weights and biases, respectively.
(8)$$Q_{d}=\underset{Q}{argmin}\sum |W_{float}^{l}-W_{fixed}^{l}\left(w,Q\right)|.$$
(9)$$Q_{d}=\underset{Q}{argmin}\sum |b_{float}^{l}-b_{fixed}^{l}\left(w,Q\right)|.$$

In the stage of quantization of inputs and outputs between layers, we find the optimal *Q* value for each layer of the input–output feature map, and the optimal *Q* value is calculated as shown in Equations (10) and (11). For example, the RGB value of the input image is scaled to the [0,1] interval in the preprocessing stage, and *Q* = 14 can be used to quantize the input of the first layer when the bit width *w* = 16.
(10)$$Q_{d}=\underset{Q}{argmin}\sum |I_{float}^{l}-I_{fixed}^{l}\left(w,Q\right)|.$$
(11)$$Q_{d}=\underset{Q}{argmin}\sum |O_{float}^{l}-O_{fixed}^{l}\left(w,Q\right)|.$$

In the stage of quantifying the intermediate results, we find the best *Q* value for each layer of intermediate data by using the approach shown in (12).
(12)$$Q_{d}=\underset{Q}{argmin}\sum |Inter_{float}^{l}-Inter_{fixed}^{l}\left(w,Q\right)|.$$
By quantifying in the above way, the model size can be further reduced to 50% after pruning, reducing the consumption of computing, memory, and bandwidth resources.

### 3.2. Self-Generated REID-UADETRAC Dataset

The Deepsort algorithm uses a small CNN to extract the appearance features of the detected target. For each detected frame, the similarity of the detected target in the current frame is compared with the appearance features previously saved. The original network is used to extract the appearance features of pedestrians; thus, the default width × height of the input is 64×128. To ensure no distortion of the input feature information, we changed it to 128×128 to better conform to the size of vehicles in the monitoring video stream.

To adapt the Deepsort algorithm to vehicle tracking, we need a large dataset to train the vehicle appearance feature extraction network. Currently, there are few reidentification datasets based on the roadside traffic monitoring of vehicles. Therefore, based on the vehicle recognition information and detection frame information provided by UA-DETRAC, we generated the vehicle reidentification dataset, REID-UA-DETRAC. To ensure the effectiveness of training, we selected vehicles that appeared more than 200 times as valid data. The generated dataset contained 1053 identities and 606,007 images in total. Part of the generated dataset is shown in Figure 4.

### 3.3. Overview of the Accelerator Architecture

Based on the YOLOv3 and YOLOv3-tiny models, we designed the architecture of the vehicle detection accelerator. All kinds of calculations required by the model are optimized and encapsulated into the corresponding calculation modules in the FPGA. The overall architecture of the accelerator is shown in Figure 5.

The accelerator consists of a host computer and an FPGA. The main tasks of the host are image preprocessing, data quantization, nonmaximal suppression, and Deepsort task scheduling. The host uses the controller to schedule the flow of data and uses the memory manager to manage the interaction between DRAM and DMA. The FPGA is responsible for the accelerated calculation of various computation-intensive tasks. The host loads the configuration information of the current model at the beginning, and stores the pre-quantized weight and bias data of the model in a continuous memory. Then, the host extracts the input video into frame images and sends them to the controller module in sequence. First, the control module sends the image to the data quantification module. Then, it transfers the quantized image, the weight, and bias data of the current layer to the FPGA on-chip memory through the optimized transmission method of ping-pong double buffering and multi-channel transmission. After the acceleration of a specific computing module, the result is sent back to the off-chip DRAM through the above method. After completing the prediction of an image, the host performs the NMS (Non Maximum Suppression) operation and transmits the result to the Deepsort tracking module. Finally, it draws the tracking result into a new video stream in real time.

### 3.4. Strategies of Memory Optimization

DNNs contain a large number of parameters; however, the on-chip Block Random Access Memory (BRAM) cannot carry all the data. Storing all the parameters in the off-chip Dynamic Random Access Memory (DRAM) will significantly increase the access latency. Therefore, we use DRAM to store image data and weight data. During the operation, the data blocks involved in the operation are transmitted to BRAM, and the results are written back to DRAM after the operation is completed.

#### 3.4.1. Model Configurability and Memory Interlayer Multiplexing

The accelerator we proposed encapsulates the operators required by the YOLOv3 and YOLOv3-tiny models. In the stage of designing the overall architecture, we take into account the configurability of the accelerator, and design two memory management drivers for the two models, respectively. Such a design can switch detection models by adaptively or manually switching memory management drivers based on real-time traffic and weather conditions. When the detection environment is poor, using YOLOv3 can achieve a higher accuracy. However, when the traffic increases, the calculation using YOLOv3 takes a long time, and the real-time detection effect can be achieved by using YOLOv3-tiny. Based on this, when the traffic flow is high (the detection results of 10 consecutive frames are more than 15 vehicles), the software automatically switches to use the memory management driver of YOLOv3-tiny to improve the detection speed. When the detection environment is not ideal (rainy days, night scenes), we can manually switch to the YOLOv3 model to obtain a higher detection accuracy. We read the parameters of both models into the specified memory in the program loading stage, and the switching action is after the end of a picture detection, so there is almost no switching delay. According to the accuracy priority and speed priority, the accelerator can better adapt to the change in the detection environment. The following is an introduction to the two memory driver designs.

Since the input of YOLOv3 and YOLOv3-tiny is the output of the previous layer except for the first layer, we use the strategy of memory interlayer multiplexing to reduce DRAM consumption.

Figure 6a illustrates the intermemory multiplexing strategy adopted by YOLOv3-tiny. The memory management module of YOLOv3-tiny uses two caches of the same size, each 416×416×3, which is the largest feature map of all layers. Typically, outputs in adjacent layers are written to the head of the first cache and the tail of the second cache. The Route layer needs to stitch the results of two layers. For example, the output of YOLOv3-tiny’s Layer(20) is the stitched result of Layer(8) and Layer(19) outputs; thus, the output of the middle layer needs to be sequentially cached at the position indicated by top1 until the results of Layer(8) and Layer(19) are stitched at the bottom of the second cache, as shown in Figure 6b, and used as Layer(20)’s output.

The interlayer multiplexing strategy adopted by YOLOv3 is the same as that of YOLOv3-tiny. Since the YOLOv3 network has more layers and a more complex model, its memory management module consists of five caches of size 416×416×3.

#### 3.4.2. Parameter Rearrangement in Memory

The convolution operation requires the parallel calculation of the results of all channels and then summing them. The original weight parameters are stored in memory in row priority order, which requires more access times. According to the block division of the weight of each layer, we reordered the weight parameters of each layer in advance, which reduced the accessing delay time during the operation [37]. The parameter conventions are shown in Table 2.

Taking the weight parameters of the 12th layer of YOLOv3-tiny as an example, as shown in Figure 7, there are $1024\times 512\times 9(X\times Y\times K^{2})$ parameters, where X=1024,Y=512, and $K^{2}=9$. X,Y represent the number of input and output feature maps, respectively. When the convolutional loop block is divided according to Tx=32, Ty=4, and the weight parameters are stored in row priority order, 524,288 parameter blocks with a size of 9 need to be read from the memory in the order of the arrow. After rearrangement, the parameters are stored continuously, and 4096 parameter blocks with a size of 32×4×9 should be read from memory in the order of the arrows. Parameters’ prearrangement reduces memory reads.

#### 3.4.3. Multichannel Transmission

We adopt the multichannel transmission strategy shown in Figure 8 to optimize data transmission and further reduce the transmission delay.

The input characteristic diagram is transmitted by n channels in parallel, and the output characteristic diagram is transmitted by m channels in parallel. After the current layer is partitioned, each incoming channel reads the pixel blocks corresponding to $\left\lceil \frac{Tx}{n}\right\rceil$ feature maps from DRAM into the chip each time. The size of each input pixel block is trow×tcol=((tr−1)×S+K)×((tc−1)×S+K). After the operation is completed, each output channel reads $\left\lceil \frac{Ty}{m}\right\rceil$ feature image pixel blocks with size tr×tc from the BRAM and writes them out of the chip. Due to the small amount of weight parameter data, $DMA_{0}$ is used for transmission, $DMA_{1}$ to $DMA_{n}$ are used to transmit the input characteristic map, and $DMA_{n+1}$ to $DMA_{n+m}$ are used to transmit the output characteristic map.

After an exhaustive architecture search, we give the values of these parameters to determine the final accelerator architecture: $Tx_{max}=32,Ty_{max}=4,n=4,m=2$.

For the convolutional layer, Tx and Ty can be described as (13) and (14).
(13)$$Tx=min(X,Tx_{max}).$$
(14)$$Ty=min(Y,Ty_{max}).$$

The Tx and Ty of the pooling layer can be described as (15).
(15)$$Tx=Ty=min(Tx_{max},Ty_{max},X).$$

The architecture determined by the above parameter reduces the number of feature maps transmitted by each channel from Tx+Ty to $\frac{Tx}{n}$ or $\frac{Ty}{m}$ without causing too much competition, which brings us the best transmission delay.

#### 3.4.4. Multi-Level Pipeline Optimization

Due to the dependence of data transfer between two adjacent layers and the convolutional module needing to traverse the output of all channels of the previous layer, the interlayer flow optimization cannot be carried out. However, there is no data dependence among all channels in the calculation of the pooling layer, so some data from the upper layer can be used for direct calculation. Therefore, we fuse the convolution module with the pooling module to reduce the triple access of the two operations to one, as shown in Figure 9.

For the convolution module designed with the Im2col+GEMM algorithm, we use the in-layer pipeline design shown in Figure 9. The entire convolution module is optimized into a four-stage pipeline, corresponding to four subtasks: the line cache, Im2col function, GEMM calculation, and result output.

### 3.5. Strategies of Computational Optimization

#### Multiscale Convolution Acceleration Engines

The convolution operation is used for feature extraction of images, and its computational complexity is $O\left(OH\times OW\times K^{2}\times IC\times OC\right)=O\left(n^{6}\right)$. It is the most computationally resource-consuming operation in the YOLO family of networks and is our main target for acceleration. The convolution operation can be described by (16). After omitting the index symbol of the tensor operation, Equation (Equation 16) can be simplified to (17). A schematic diagram of the convolution calculation is shown in Figure 10.
(16)$$\begin{array}{cc} \; & \begin{array}{cc} O[oc][oh][ow]= & \sum_{c=0}^{C-1}\sum_{i=0}^{K-1}\sum_{j=0}^{K-1}I[ic][S*oh+i][S*ow+j]\times \\ \; & W[oc][ic][i][j]+B[oc], \end{array}\\ \; & 0\leq oc<OC,0\leq ic<IC,0\leq i,j<K\\ \; & OH=\frac{IH-K+2pad}{S}+1,\\ \; & OW=\frac{IW-K+2pad}{S}+1 \end{array}$$
(17)O=I×W+B

CNN computing requires a large amount of memory, but the FPGA’s on-chip storage resources cannot meet the requirement of storing such a large amount of data at a time [38]. Therefore, based on the local principle of the convolution computing data, the input feature map data and corresponding weight parameters can be divided into blocks. Each time, 2 pixel blocks of size Tx×tir×tic and corresponding weight parameters of size $Tx\times Ty\times K^{2}$ are read from the off-chip DRAM. After all the on-chip data are calculated, the result of size Ty×tor×toc is written back to the off-chip DRAM. The calculation of tor is shown below:(18)$$tor=\frac{tir-K+2pad}{S}+1$$

We deeply analyze the characteristics of convolution operations with kernel sizes of 1×1 and 3×3, and design two convolution acceleration engines using the Im2col+GEMM [39] and Winograd [40] algorithms, respectively, so as to reduce the computational complexity and resource consumption.

**Im2col+GEMM:** The Im2col+GEMM algorithm reduces the time complexity of the convolution operation from $O(n^{6})$ to $O(n^{3})$ by using the matrix multiplication instead of the convolution, as shown below:

The GEMM algorithm needs to stretch the convolution kernel and feature map into the matrix form. This transformation process not only includes the delay increase caused by the multiple access but also requires more storage space to temporarily store the matrix form of the feature map for which the kernel size is larger than 1×1. Figure 11 illustrates a 2D convolution operation with a 2×2 convolution kernel, which requires 4×4 space to temporarily store the matrix form of a 3×3 feature map. Without a loss of generality, for the convolution kernel with size K×K, the im2col+GEMM algorithm is used to convolve the feature map with size IH×IW. Combined with (18), the ratio of the spatial complexity to be improved is shown below:(19)$$space\_ratio=\frac{K^{2}\times OH\times OW}{IH\times IW}$$

The space complexity is proportional to the convolution kernel size $K^{2}$. The convolution kernel with a size of 1×1, as shown in Figure 12, does not consume extra space to store the feature graph matrix; thus, the im2col+GEMM algorithm is more suitable for acceleration.

Figure 13 illustrates the convolution module based on the Im2col+GEMM algorithm. DMA transfers a feature map with a size of Tx×tr×tc and corresponding weight parameters with a size of Tx×Ty to the on-chip memory each time. The input feature map is stretched into Tx vectors with length tr×tc by the Im2col function in the direction indicated by the arrow. Then, we send Tx×Ty weights and the feature map vector into a parallel multiplier for the dot product operation. After this, Tx×Ty intermediate results with a length of tr×tc are divided into Ty groups according to the output dimension and sent to the parallel adder. The additional result of each group is the result of a certain layer of the output feature graph. After all groups are operated by the adder, the final result is converted into the matrix form of Ty×tr×tc as the output.


**Winograd convolution:**


For the convolution operation with a convolution kernel size of 3×3, we designed a Winograd convolution engine to accelerate the operation.

The Winograd algorithm accelerates the convolution operation by significantly reducing the multiplication operation in the convolution [41]. F(m×m,r×r) represents a two-dimensional convolution function; its input is a convolution kernel of size r×r, and the output is an output feature map of size m×m. We use *Y* to represent the output of this function, which can be expressed in the form of Equation (Equation 20).
(20)$$Y=A^{T}\left[\left[GWG^{T}\right]⊙\left[B^{T}IB\right]\right]A$$

In Equation (Equation 22), *W* represents the convolution filter, *I* represents the input feature map, *G* is the convolution kernel transformation matrix of size r(m+r−1), *A* is the output feature map transformation matrix of size m(m+r−1), and *B* is the input feature map transformation matrix of size ${\left(m+r-1\right)}^{2}$. $A^{T},B^{T},G^{T}$ are transpose matrixs of A,B,G. The calculation of the convolution using the sliding window algorithm requires $m^{2}r^{2}$ multiplication operations, which can be reduced to (2m−1)(2r−1) operations by using the Winograd algorithm. The time complexity decreases from $O(n^{4})$ to $O(n^{2})$ when we ignore the effect of the addition operations.

Figure 14 illustrates the convolution module based on the Winograd algorithm. First, DMA passes the feature graph with size Tx×tir×tic and the corresponding weight parameters with size $Tx\times Ty\times K^{2}$ into on-chip memory. Then, we transform the input feature map and convolution kernel into the same dimension matrix and send them into the parallel multiplication tree for the multiplication operation. After that, we fed the intermediate result into the parallel addition tree after the matrix transformation to calculate the convolution result.

### 3.6. Max-Pooling and Upsampling Parallel Optimization

Max-pooling is a downsampling operation that selects the largest value in an image region as the pooled value in that region based on the stride and the pooling size [42]. Pooling compresses the scale of feature maps, reduces the computational burden of models and keeps the scale invariance and rotation invariance. The output sizes OH and OW can be calculated as follows:(21)$$\begin{array}{cc} \; & OH=OW=\lfloor \frac{IS-K}{S}\rfloor +1,\\ & IS=IH=IW. \end{array}$$

The max-pooling module is implemented by the comparator. Figure 15 illustrates the composition of its hardware architecture. When the pipeline is full, *n* pixels at the same position of the feature map are read in parallel in each clock cycle, and the pooling result is obtained after the $K^{2}$ parallel comparison operation in *n* paths and written into the output cache.

Each max-pooling module directly reads the convolution result of the previous layer from on-chip memory. They compare the read result with the predefined minimum value MIN and store the larger value in register Reg, while the counter Cnt is recorded as 1. In the next comparison, the input value and the value in the register are fed into the comparator, while the counter Cnt increases by 1 until it increases to $K^{2}$. *N* pooling modules run in parallel, and the result of *N* feature maps can be obtained after the $K^{2}+1$ clock cycle.

Upsampling is an image magnification technique that improves the image resolution by interpolation. This operation first takes every pixel in the image and makes four copies of it, doubling the width and height of the output feature image.

Because the upsampling calculation process is simple and the calculation scale is small, we use the combinatorial logic circuit to implement the module. The optimization idea we use is similar to the max-pooling module, so we will not go into detail here.

#### Fused Convolution and Batch Normalization Computation

Batch normalization is often used in DNNs to accelerate network convergence after linear calculation and before nonlinear activation [43]. To reduce the computation, we merge the BN operation into the convolution process. Equation (Equation 22) rearranges (17) to achieve the BN of the convolution results. The parameter conventions are shown in Table 3.
(22)$$O_{norm}=\frac{\gamma }{\sqrt{\sigma^{2}+\epsilon }}O+\left(\frac{\gamma \mu }{\sqrt{\sigma^{2}+\epsilon }}+\beta \right).$$

Let $P=\frac{\gamma }{\sqrt{\sigma^{2}+\epsilon }}$ and $Q=(\frac{\gamma \mu }{\sqrt{\sigma^{2}+\epsilon }}+\beta )$; then, Equation (Equation 22) can be simplified into the form in (23).
(23)$$O_{norm}=PO+Q.$$

Substituting (17) into (23), we obtain (24).
(24)$$O_{norm}=P\left(I\times W+B\right)+Q.$$

Equation (Equation 22) is rearranged to obtain $W_{new}=PW$ and $B_{new}=PB+Q$. Then, $O_{norm}$ can be described, as shown below:(25)$$O_{norm}=I\times W_{new}+Bnew.$$

By integrating BN with a convolution operation, computing resources and delay can be reduced.

## 4. Experiments

### 4.1. Experimental Setup

We designed and simulated the proposed accelerator to verify the effectiveness of the proposed optimization method. The training and pruning quantization were completed by the NVIDIA Tesla V100 platform. The detection inference was implemented by the CPU+FPGA heterogeneous platform. The chip we used is ZYNQ XC7Z035-FFG676-2. We designed the IP cores of YOLOv3 and YOLOv3-tiny accelerators using Xilinx Vivado HLS 2021.2, and used Vivado 2021.2 for the synthesis and layout.

### 4.2. Dataset and Model Training

We used the UA-DETTAC dataset to train the YOLOv3 and YOLOv3-tiny detection models. It is a large-scale dataset used for vehicle detection and tracking, covering 1.21 million detection frames of 8250 vehicles under different weather and road conditions.

During training, the four categories of the dataset were merged into a single category, called car. The same training parameters were selected in the training stages of the two models. After basic training and pruning for the YOLOv3 and YOLOv3-tiny models, the specific performance indices are shown in Table 4.

According to the data in Table 4, the size of the pruned YOLOv3 is reduced by 85%. The detection accuracy AP@0.5 is improved by 0.04, and the number of floating-point calculations required for convolution is reduced by 70.4%. The size of YOLOv3-tiny after two dynamic prunings reduced 98.2% at the cost of the mAP reduction by 0.026. The computation of the convolution is reduced by 86.5%.

To show the detection effect more intuitively, we selected an image from the test set to be detected using the models before and after compression, as shown in Figure 16.

The number of vehicles detected by each model is shown in Table 5. Among them, the detection results of the YOLOv3-prune 85% model are consistent with those of the original YOLOv3 model, both detecting 27 vehicles. The detection accuracy of the YOLOv3-prune 85% model is slightly better than that of the original YOLOv3 model. The detection performance of the YOLOv3-tiny-prune 85% + 30% model is slightly better than that of the original YOLOv3-tiny model. The number of vehicles is increased from 24 to 26.

### 4.3. RE-ID Deepsort

The UA-DETRAC dataset is derived from the road surveillance video in real scenarios. Therefore, the images in the test set can be converted to the video with a frame rate of 24 fps for the tracking test of the Deepsort algorithm. A comparison of the tracking results before and after training using the reidentification dataset is shown in Figure 17.

We further used the MOT-Metrics tool [44], combined with identification precision (IDP), identification recall (IDR), the corresponding F1 score IDF1, false negative (FN), false-positive (FP), ID switch (IDs), multiobject tracking accuracy (MOTA), and multiobject tracking precision (MOTP) metrics, to measure the tracking performance. The upper arrow ↑ means the larger value shows better performance, and the lower arrow ↓ means the smaller value shows better performance. We test using three video streams from the UA-DETRAC test set, MVI_40701, MVI_40771, and MVI_40863, which are challenging and cover a variety of traffic conditions during the peak traffic flow in daytime and nighttime and on rainy days. The results of the ablation experiments are shown in Table 6.

The MVI_40701 video stream was captured in the daytime peak traffic flow scene and shot from a forward overlooking angle. All evaluation indices show that the RE-ID Deepsort algorithm has better performance than the Deepsort algorithm in tracking vehicles.

The MVI_40771 video stream was captured in the peak traffic flow scene at night and shot from a forward overlooking angle. Compared with the Deepsort algorithm, the RE-ID Deepsort algorithm improved significantly in several evaluation indices, especially in reducing the number of IDs. The experimental results indicate that the model proposed in this paper has an obvious improvement effect on vehicle detection in night scenes.

The MVI_40771 video stream was shot on a rainy day with heavy traffic. The shooting angle was a side view. A large number of small cars were covered by large cars in this video stream; thus, all tracking indices are inferior to those of the previous two video streams. Except for the same MOTP, RE-ID Deepsort is better than Deepsort in other indicators.

We use the YOLOv3 and YOLOv3-tiny models after pruning and combine them with the RE-ID Deepsort algorithm to conduct vehicle tracking counting experiments. Figure 18 is the result of vehicle detection and ID assignment. When a vehicle crosses the solid red line in the figure, one is added to the traffic flow counter. Figure 19 compares the traffic flow data collected by the model with the data collected by manual statistics.

The scenarios we tested included peak traffic under the day, night, and rainy conditions. In these scenarios, vehicles move slowly, and the vehicles are relatively close, which easily generates occlusion. A car located between two large vehicles will cause missed detection. Especially in the third scenario, the camera is located on the side of the road; thus, cars in the middle of the road are almost completely covered by large cars on the side road when moving slowly, resulting in poor detection results. The accuracy rates of YOLOv3 and YOLOv3-tiny were 96.15% and 92.3% in daytime conditions, 94.0% and 92.0% in nighttime conditions, and 81.8% and 75.8% in rainy conditions, respectively.

### 4.4. Comparison and Discussion

Table 7 shows our cross-platform comparison results. We implement the same YOLOv3-tiny network in CPU and GPU, then we measure the performance using frame per second and power consumption (W). Our proposed accelerator design has a 33× higher energy efficiency ratio and 9.16× higher forward inference speed than the AMD Ryzen 7 5800H CPU. Compared to the NVIDIA GeForce RTX 2060, our FPGA implementation achieves similar throughput while achieving a 9.4× improvement in energy efficiency. We take the acceleration results based on Xilinx Vitis AI and DPU as another baseline. The ZCU102 evaluation board uses the mid-range ZU9 UltraScale+ device. Compared with the device we use, it has more abundant resources and better performance. However, the price is more expensive; our design with the Zynq-7000 outperforms it in terms of cost efficiency and energy efficiency. Compared with the yolov3_adas_pruned_0_9 model implemented based on Vitis AI and ZCU102, our pruned YOLOv3-tiny model has faster forward inference speed and higher fps. Tajar and his partner implement the YOLOv3-tiny network for the vehicle detection on Nvidia Jetson Nano [45]. The throughput of their solution was not sufficient for applications requiring at least 24 fps. However, we achieve 91.65 fps with the pruned YOLOv3-tiny model. We compare the cost efficiency of different platforms, and the results demonstrate that our solution is the most cost-effective.

Table 8 shows the comparison of our work with previous fpga-based work. Since we designed pipeline processing based on both intralayer and interlayer granularity, the computational efficiency is slightly higher than that of the literature [14]. The resource consumption is slightly higher than those in the literature [14] because of the separate upsampling computation module we designed to adapt the computation of the upsampling layer in YOLOv3 and YOLOv3-tiny. However, due to our pruning strategy, we reduced the computation of YOLOv3 and YOLOv3-tiny by factors of 3.4 and 7.4, respectively. Our design has a significant increase in throughput, with a slightly better computational performance. We doubled the performance of the convolutional computation compared to the literature [37] due to the introduction of the Winograd convolutional acceleration computation engine and multiple levels of pipeline processing. Since the literature [14,37] only shows dynamic power consumption, for a fair comparison, we use dynamic energy efficiency (GOPS/W) to compare with them and obtain a clear advantage. At the same time, we also have better cost efficiency and DSP efficiency. Reference [33] uses lower bit quantization precision and introduces a new sparse convolution algorithm, which makes the DSP efficiency higher. They use an end-to-end design space search for a sparse convolution-specific circuit architecture, making it computationally more efficient than our design. Since our YOLOv3-tiny model is less computationally intensive after compression, we have a higher detection speed. At the same time, our designs consume less power and are more cost-effective. Ding et al. [46] proposed a resource-aware system-level quantization framework, which takes into account both the accuracy of the object detection algorithm and the hardware resource consumption during deployment. They implemented the acceleration of the YOLOv2-tiny network on the Virtex-7 with more abundant resources and superior performance, and achieved a high throughput. Our design deploys the more advanced YOLOv3 and YOLOv3-tiny models at less than 10% of the overall resource consumption of their design, and outperforms it in terms of the DSP efficiency. None of the models in other works are trained on vehicle detection-specific datasets; thus, our model has an advantage in vehicle detection scenarios.

### 4.5. Scalability Discussion

To cope with the demand of detecting multiple video streams at one intersection in practical applications, we implemented a design of deploying multiple accelerator modules on-chip xc-7z035-ffg676-2. The experimental results shown in Table 9 show that the detection throughput of the Fixed-16 precision YOLOv3-tiny model reaches 168.72 fps, which satisfies the demand for the simultaneous detection of up to six real-time video streams with a frame rate of 25 fps at the same intersection.

Our acceleration architecture currently supports the acceleration of three operators: convolution, upsampling, and max-pooling. Many auxiliary functions come from darknet, so it has good support for YOLO series networks. It also supports YOLOv2 and YOLOv2-tiny. Since their detection performance is not as good as YOLOv3 and YOLOv3-tiny, we do not compress them. The inference speeds of the original YOLOv2 and YOLOv2-tiny are 433 ms and 72 ms, respectively. It can support YOLOv4 network by adapting the memory management driver. In order to adapt to YOLOv5, the focus structure needs to be optimized.

By adapting memory management drivers and auxiliary functions such as load_network and get_detections, our acceleration architecture can be applied to other networks such as SSD, ResNet, and MobileNet.

## 5. Conclusions and Future Work

We implemented a scalable vehicle detector based on the FPGA and the YOLO object detection algorithm using HLS. In terms of model compression, we incorporate the dynamic threshold structured pruning strategy based on the binary search and the dynamic INT16 fixed-point quantization algorithm to significantly reduce the model size and computation. In terms of vehicle tracking, we generated a reidentification dataset based on the UA-DETRAC dataset and used it to train the appearance feature extraction network of the Deepsort algorithm with a modified input size to improve the vehicle tracking performance. For the two convolution operations, we designed fast convolution engines based on the Winograd and GEMM algorithms, respectively. For max-pooling and upsampling computation in YOLOv3 and YOLOv3-tiny models, we use combinatorial logic to implement parallel computation to improve the computational efficiency.

Compression acceleration of neural networks still has many worthy research directions. In the future, the search for network structures suitable for solving specific problems using the optimization scheme of neural architecture searches (NASs) can be used as a research direction for network compression. In terms of hardware acceleration, the current neural network accelerators usually only support a few network frames, and the generalization of acceleration frames is also worthy of further research.

## Figures and Tables

**Figure 1 sensors-23-02208-f001:**
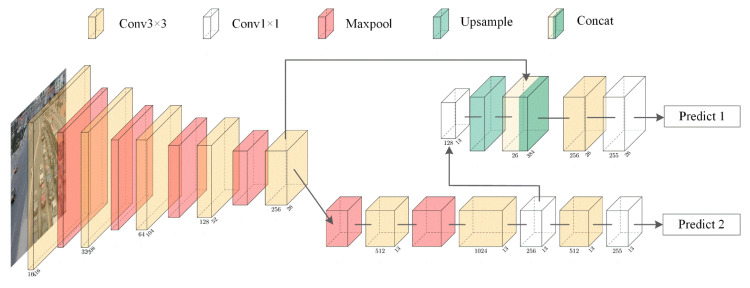
Network structure of YOLOv3-tiny.

**Figure 2 sensors-23-02208-f002:**
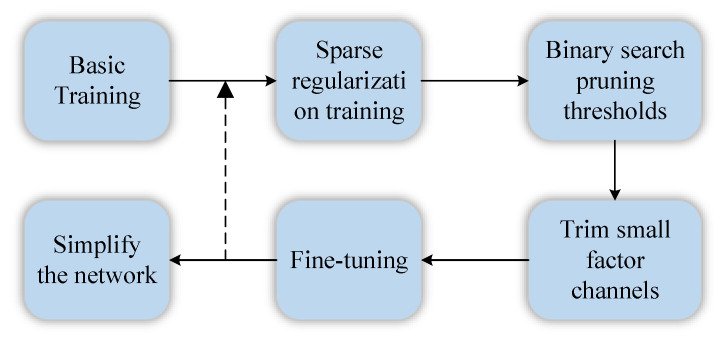
Strategy of dynamic threshold pruning.

**Figure 3 sensors-23-02208-f003:**
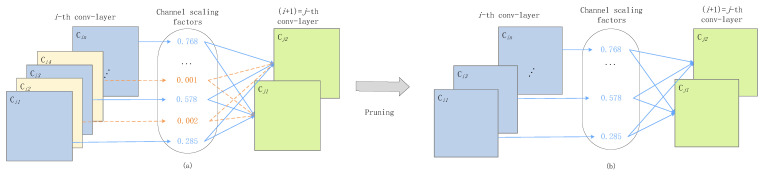
Strategies for sparse regularized channel pruning. (**a**) Structure before pruning. (**b**) Structure after pruning.

**Figure 4 sensors-23-02208-f004:**
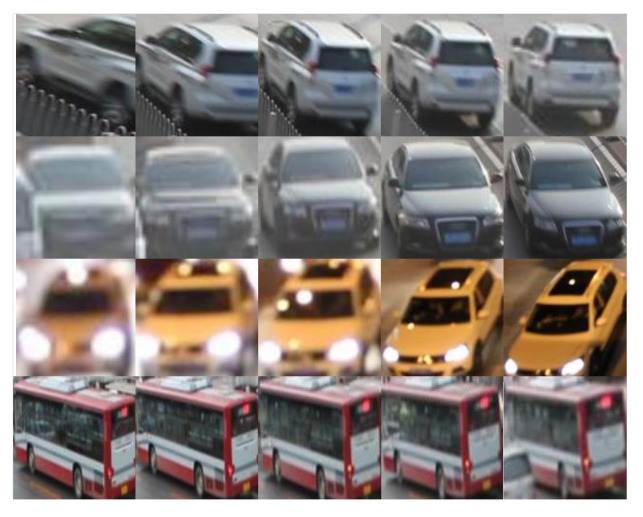
Images from the REID-UADETRAC dataset.

**Figure 5 sensors-23-02208-f005:**
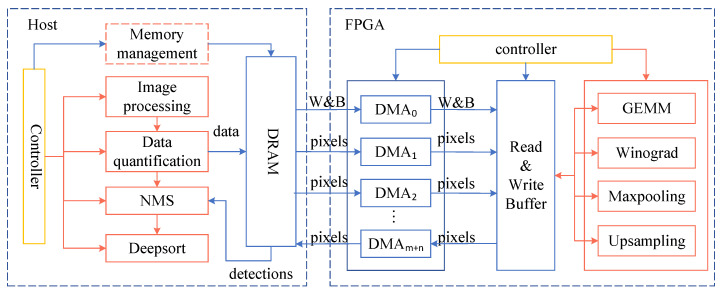
Overview architecture of the accelerator.

**Figure 6 sensors-23-02208-f006:**
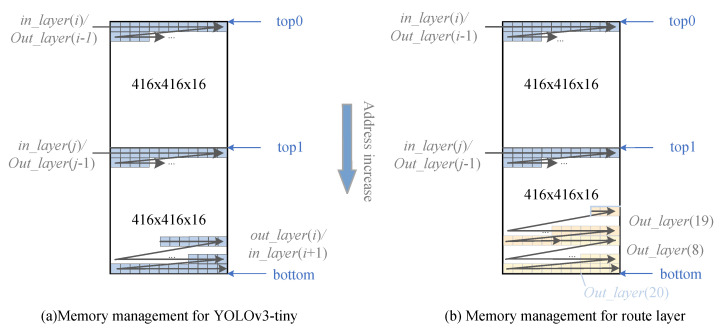
Diagram of the memory interlayer multiplexing strategy.

**Figure 7 sensors-23-02208-f007:**
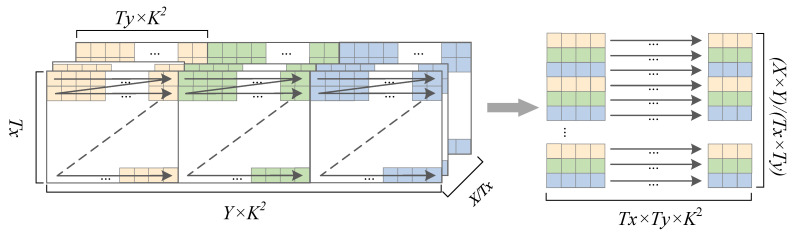
Parameter rearrangement.

**Figure 8 sensors-23-02208-f008:**
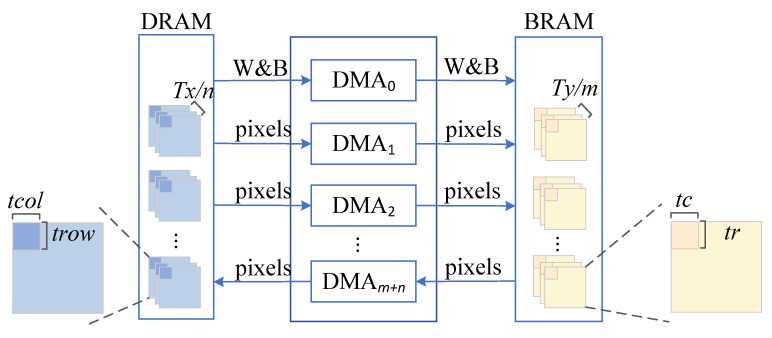
Multichannel transmission.

**Figure 9 sensors-23-02208-f009:**
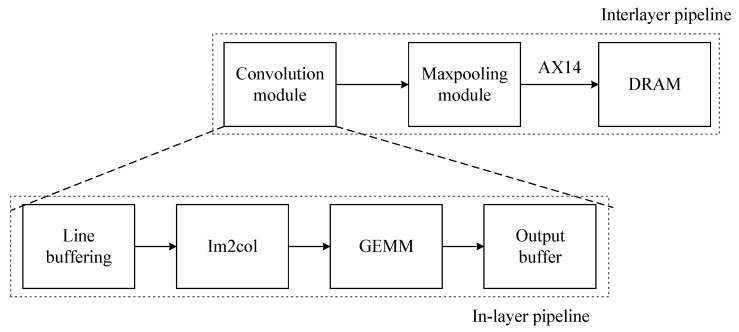
Pipeline processing.

**Figure 10 sensors-23-02208-f010:**
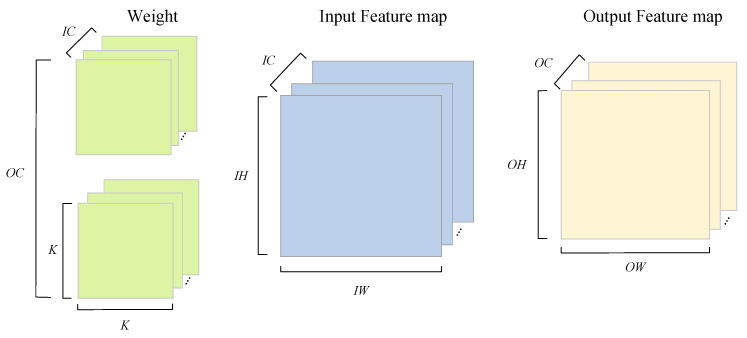
Convolutional calculation.

**Figure 11 sensors-23-02208-f011:**
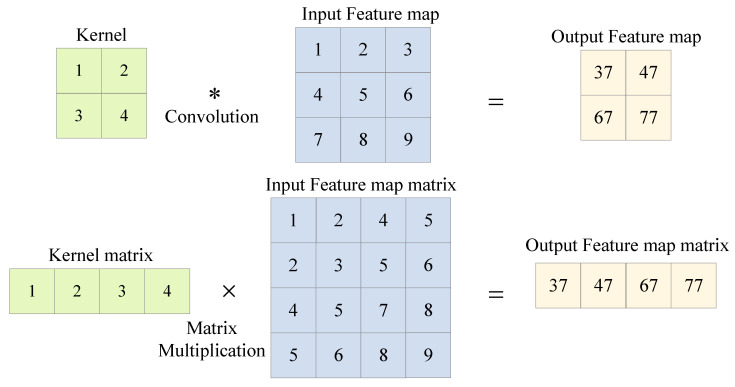
Two-dimensional convolution with kernel size 2×2.

**Figure 12 sensors-23-02208-f012:**
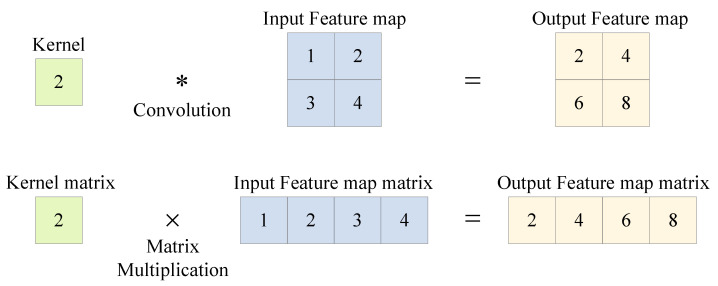
Two-dimensional convolution with kernel size 1×1.

**Figure 13 sensors-23-02208-f013:**
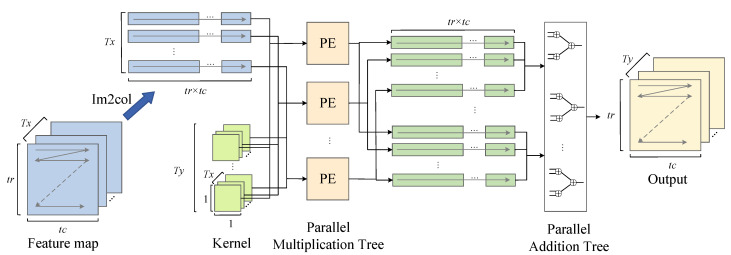
Architecture of the convolution module based on the Im2col+GEMM algorithm.

**Figure 14 sensors-23-02208-f014:**
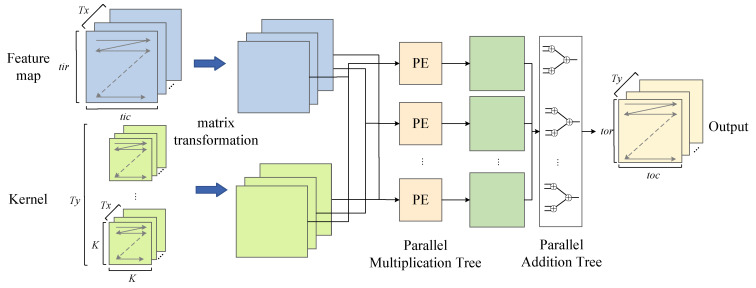
Architecture of the convolution module based on the Winograd algorithm.

**Figure 15 sensors-23-02208-f015:**
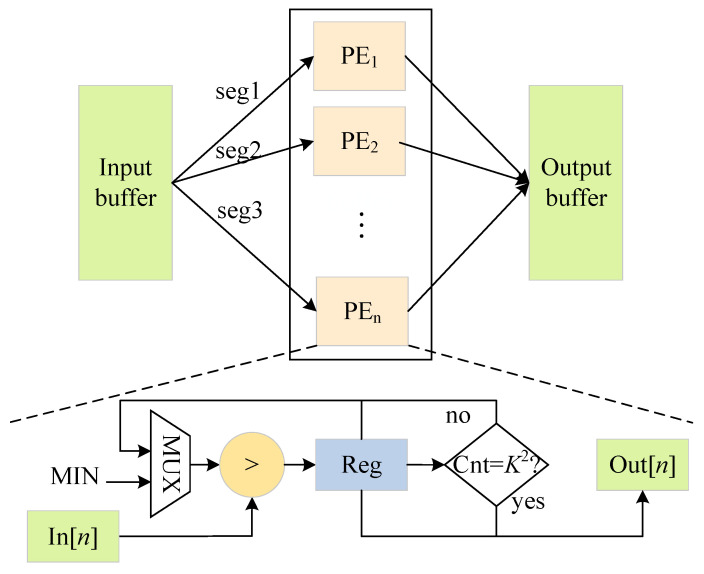
Architecture of the max-pooling module.

**Figure 16 sensors-23-02208-f016:**
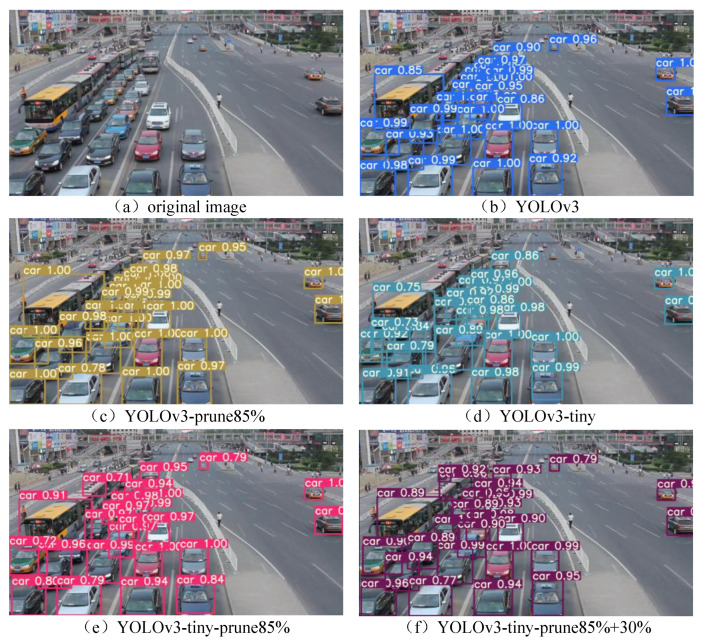
Comparison of detection results.

**Figure 17 sensors-23-02208-f017:**
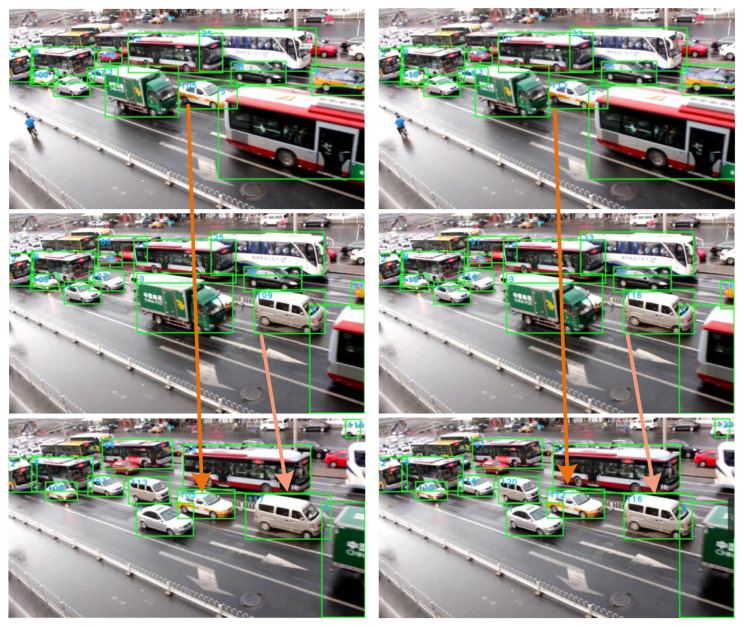
Comparison of tracking results before and after reidentification training.

**Figure 18 sensors-23-02208-f018:**
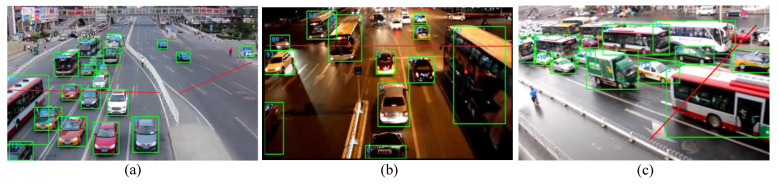
Tracking using RE-ID Deepsort. (**a**) Detection result of MVI_40701. (**b**) Detection result of MVI_40771. (**c**) Detection result of MVI_40863.

**Figure 19 sensors-23-02208-f019:**
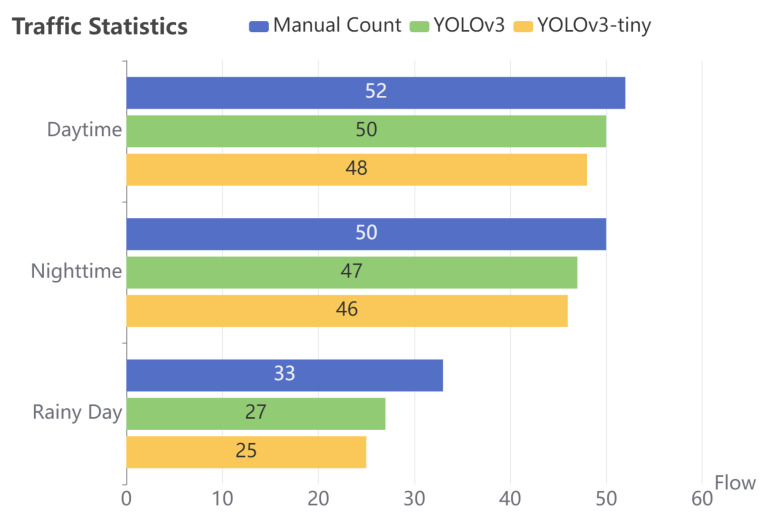
Comparison of traffic flow statistics.

**Table 1 sensors-23-02208-t001:** Meanings of the variables.

Variables	Meaning
λ	The parameter for regulating the effect of the spatial Mahalanobis distance and visual distance on the cost function.
$y_{i}$	The state vector of the *i*-th prediction frame.
$S_{i}$	The covariance matrix of the average tracking results between the detection frame and track *i*.
$d_{j}$	Detection box *j*.
$r_{i}$	The appearance descriptor extracted from detection box *j*.
$R_{i}$	The last 100 appearance descriptor sets associated with track *i*.

**Table 2 sensors-23-02208-t002:** Parameter conventions for convolution operations.

Symbol	Meaning
*I*	The input feature map.
*W*	The weights of the convolution layer.
*B*	The bias of the convolution layer.
*O*	The output feature map.
IH	The height of the input feature map.
IW	The width of the input feature map.
IC	The number of input channels.
*K*	The kernel size.
OH	The height of the output feature map.
OW	The width of the output feature map.
OC	The number of output channels.
pad	The padding.
*S*	The stride.
Tx	Parallelism of multiply-add operations on input feature maps.
Ty	Parallelism of multiply-add operations on output feature maps.

**Table 3 sensors-23-02208-t003:** Parameter conventions.

Symbol	Meaning
$O_{norm}$	The output of the feature map after batch normalization.
γ	The parameter that controls the variance of $O_{norm}$.
$\sigma^{2}$	The variance of *O*.
ϵ	A small constant used to prevent numerical error.
*O*	The output of the feature map.
μ	The estimate of the mean of *O*.
β	The parameters that control the mean of $O_{norm}$.

**Table 4 sensors-23-02208-t004:** Meanings of the variables.

Model	Pruning Rate	AP@0.5	Model Size (MB)	Parameters ($\times 10^{3}$)	BFLOPs
YOLOv3	0	0.671	235.06	61523	65.864
YOLOv3	85%	0.711	33.55	8719	19.494
YOLOv3-tiny	0	0.625	33.10	8670	5.444
YOLOv3-tiny	85%	0.625	1.02	267	1.402
YOLOv3-tiny	85% + 30%	0.599	0.59	69	0.735

**Table 5 sensors-23-02208-t005:** Comparison of the numbers of detected vehicles.

Model	Number of Vehicles Detected
YOLOv3	27
YOLOv3-prune85%	27
YOLOv3-tiny	24
YOLOv3-tiny-prune85%	24
YOLOv3-tiny-prune85%+30%	26

**Table 6 sensors-23-02208-t006:** Ablation experiment of re-id training.

Model	Video Stream	IDF1↑	IDP↑	IDR↑	FP↓	FN↓	IDs↓	MOTA↑	MOTP↓
Deepsort	MVI_40701	76.4%	82.5%	71.1%	1515	3706	53	66.8%	0.118
RE-ID Deepsort		79.6%	86.4%	76.2%	1452	3686	27	67.5%	0.117
Deepsort	MVI_40771	69.3%	74.5%	66.2%	2409	2409	49	65.2%	0.153
RE-ID Deepsort		80.6%	87.2%	75.0%	1015	2348	13	69.6%	0.155
Deepsort	MVI_40863	55.2%	80.6%	42.0%	2076	17746	51	39.2%	0.138
RE-ID Deepsort		56.1%	82.0%	42.6%	2037	17382	35	40.5%	0.138

**Table 7 sensors-23-02208-t007:** Cross-platform comparison.

Item	Platform	CNN Model	Operation (GOP)	Throughput (fps)	Full Power (W)	Efficiency (GOPS/W)	Cost Efficiency (GOPS/$$\times 10^{2}$)
Baseline1	CPU AMD R75800H	YOLOv3-tiny	0.735	10.01	45	0.16	1.96
Baseline2	GeForce RTX 2060	YOLOv3-tiny	0.735	112.87	160	0.52	16.58
Baseline3	XCZU9EG-FFVB1156	yolov3-adas-pruned-0.9	5.5	84.1	-	3.71	4.16
Ref [45]	Nvidia Jetson Nano	YOLOv3-tiny	1.81	17	10	3.08	24.62
This work	Zynq-7000	YOLOv3-tiny	0.735	91.65	12.51	5.43	46.51

**Table 8 sensors-23-02208-t008:** Comparison with previous fpga-based work.

Item	Ref [14]	Ref [37]		Ref [33]	Ref [46]	This Work			
Basic information introduction									
Platform	ZYNQ XC7Z020	Zedboard		Arria-10GX1150	Virtex-7: XC7VX690T-2	Zynq-7000			
Precision	Fixed-16	Fixed-16		Int8	Float-32	Float-32		Fixed-16	
CNN Model	YOLOv2	YOLOv2	YOLOv2-tiny	YOLOv2	YOLOv2-tiny	YOLOv3	YOLOv3-tiny	YOLOv3	YOLOv3-tiny
Dataset	COCO	COCO		VOC	VOC	UA-DETRAC			
Hardware resource consumption									
BRAM	87.5	88		96%	1320	98.5 (19.7%)		132.5 (26.5%)	
DSPs	150	153		6%	3456	301 (33.8%)		144 (16.2%)	
LUTs	36 576	37 342		45%	637 560	38 336 (22.3%)		38 228 (22.2%)	
FFs	43 940	35 785			717 660	62 988 (18.3%)		42 853 (12.5%)	
Performance comparison									
mAP	0.481	0.481	-	0.744	0.548	0.711	0.599	0.711	0.599
Operations (GOP)	29.47	29.47	5.14	4.2	1.24	19.494	0.735	19.494	0.735
Freq (MHz)	150	150		204	200	210		230	
Performance (GOP/s)	64.91	30.15	21.97	182.36	389.90	41.39	43.47	63.51	67.91
Throughput(fps)	2.20	1.02	4.27	61.90	314.2	2.12	59.14	3.23	91.65
Efficiency comparison									
Cost Efficiency (GOPS/$×$10^{2}$)	44.45	20.65	15.05	17.49	46.75	28.35	29.77	43.50	46.51
DSP Efficiency (GOPS/DSPs)	0.433	0.197	0.144	2.004	0.113	0.138	0.144	0.441	0.472
Dynamic Power (W)	1.4	1.2	0.83	-	-	1.80	1.48	1.52	1.31
Full Power (W)	-	-	-	26	21	13.29	12.92	12.73	12.51
Dynamic Energy Efficiency (GOPS/W)	46.36	25.13	26.47	-	-	22.99	29.37	41.78	51.84
Full Energy Efficiency (GOPS/W)	-	-	-	7.01	18.57	3.11	3.36	4.99	5.43

**Table 9 sensors-23-02208-t009:** Performance and power consumption of the dual.

Precision	Float-32		Fixed-16	
Platform	Zynq-7000			
Freq (MHz)	200		209	
BRAM	230.5		263	
DSPs	602		294	
LUTs	89,014		91,108	
FFs	149,259		89,148	
CNN Model	YOLOv3	YOLOv3-tiny	YOLOv3	YOLOv3-tiny
Performance (GOP/S)	78.84	82.80	115.96	124.01
Throughput (fps)	4.04	112.66	5.95	168.72
Full Power (W)	16.72	16.06	15.64	15.18

## Data Availability

Data sharing not applicable.

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
