# Peer review of "FPGA-Based Vehicle Detection and Tracking Accelerator"

_sensors, 2023, doi:10.3390/s23042208_

Round 1
Reviewer 1 Report
The paper is well written. Authors propose many optimizations for both CNN itself and for hardware accelerator. However proposed optimizations are described in very general way. Although the results presented show that proposed solution performs well, it would be good to discuss proposed optimizations in more details.
Some suggestions:
Fig. 5 The number of DMAs is m+n. How large can this number be in practice and how does it affect implementation parameters.
Equation (13) is not sufficiently explained and it is not clear how it affects architecture design decisions. Some parameters are not included in Tab.2. correct oW -> OW. I'm not sure if this is completely correct because the indexes c, r, s are not reflected in the body of the equation.
Equation (16) spaceratio - is the subscript intended?
Equation (17) if it expresses a feature map F, where is F in the equation?
Table 8 is difficult to read. It should be reformatted and perhaps divided into two tables.
Minor spelling errors and formatting can be corrected.
Reviewer 2 Report
In this article, the authors presented the FPGA-based vehicle detector using YOLO and a deep sort algorithm. The accelerator architecture was well explained, along with the memory optimization techniques used in the implementation. A detailed background survey was performed, and the relevant references were cited. The mathematical expressions of different neural network layers used in the design were explained. A comparative result analysis is presented.
It would have been nice to see a section on HLS concepts used in the implementation.
Reviewer 3 Report
I really enjoyed reading of this work and its overall quality. However, I have some little amendments that you can find in the attached file.
